# Identification of Gene Biomarkers for Tigilanol Tiglate Content in *Fontainea picrosperma*

**DOI:** 10.3390/molecules27133980

**Published:** 2022-06-21

**Authors:** Shahida A Mitu, Praphaporn Stewart, Trong D Tran, Paul W Reddell, Scott F Cummins, Steven M. Ogbourne

**Affiliations:** 1Centre for Bioinnovation, University of the Sunshine Coast, Maroochydore DC, QLD 4558, Australia; s_m169@student.usc.edu.au (S.A.M.); ttran1@usc.edu.au (T.D.T.); scummins@usc.edu.au (S.F.C.); 2School of Science, Technology and Engineering, University of the Sunshine Coast, Maroochydore DC, QLD 4558, Australia; pstewart@usc.edu.au; 3EcoBiotics Ltd., Yungaburra, QLD 4884, Australia; paul.reddell@ecobiotics.com.au

**Keywords:** tigilanol tiglate, anti-cancer, biosynthesis pathway, gene expression, CYP94C1, R protein, ERD-7.1, 2-ARD, stress, drought

## Abstract

Tigilanol tiglate (EBC-46) is a small-molecule natural product under development for the treatment of cancers in humans and companion animals. The drug is currently produced by purification from the Australian rainforest tree *Fontainea picrosperma* (Euphorbiaceae). As part of a selective-breeding program to increase EBC-46 yield from *F. picrosperma* plantations, we investigated potential gene biomarkers associated with biosynthesis of EBC-46. Initially, we identified individual plants that were either high (>0.039%) or low EBC-46 (<0.008%) producers, then assessed their differentially expressed genes within the leaves and roots of these two groups by quantitative RNA sequencing. Compared to low EBC-46 producers, high-EBC-46-producing plants were found to have 145 upregulated genes and 101 downregulated genes in leaves and 53 upregulated genes and 82 downregulated genes in roots. Most of these genes were functionally associated with defence, transport, and biosynthesis. Genes identified as expressed exclusively in either the high or low EBC-46-producing plants were further validated by quantitative PCR, showing that *cytochrome P450 94C1* in leaves and *early response dehydration 7.1* and *2**-alkenal reductase* in roots were consistently and significantly upregulated in high-EBC-46 producers. In summary, this study has identified biomarker genes that may be used in the selective breeding of *F. picrosperma*.

## 1. Introduction

Terpenoids, also known as isoprenoids, are the largest and most diverse class of plant natural products, many of which have substantial pharmacological activity. A wide variety of terpenoids, ranging from monoterpenoids, sesquiterpenoids, and diterpenoids to triterpenoids and steroids are known from members of the Euphorbiaceae family [1,2,3], attracting considerable attention due to their valuable biological activities relating to the diterpenoid backbones and their diverse array of aliphatic and aromatic ester groups [4]. In particular, tigliane and related diterpenoids are of special interest because they have a very limited known distribution in nature, being found only in the plant families Euphorbiaceae and Thymelaeaceae [2]. The plant family Euphorbiaceae is a rich source of terpenoid natural products [5] and is particularly noted for the occurrence of diverse diterpenoid backbones supporting an array of aliphatic and aromatic ester groups [6]. Tigilanol tiglate (EBC-46) is one such terpenoid, a novel epoxy-tigliane diterpene ester discovered from the fruit of the native Australian rainforest tree *F. picrosperma* [7]. *Fontainea picrosperma* C.T. White (Euphorbiaceae) is a dioecious, subcanopy rainforest tree endemic to the Atherton Tablelands, Queensland, Australia. The species is locally common but has a restricted natural range [8,9].

EBC-46 is a novel epoxy-tigliane diterpene ester extracted from the fruit of *F. picrosperma* [7,10]. TT is likely present in all tissues of the plant, and biosynthesis occurs in the roots [11]. It is of particular interest due to its effectiveness as a local treatment for a range of cancers in humans and animals [7,12,13]. EBC-46 was approved for use as a local intratumoural therapy for canine mast cell tumours by major regulatory authorities, including the European Medicines Agency and the United States Food and Drug Administration in 2020 [10,14] and is currently in clinical development for a range of cancers in humans and in companion animals [12,15]. EBC-46 is not amenable to synthesis on a commercial scale and is currently produced by isolation and purification from the fruit of *F. picrosperma* [7,11,16]. Consequently, there is considerable interest in the potential for selective breeding of *F. picrosperma* to improve yields of EBC-46.

At present, neither the site nor the pathway leading to the biosynthesis of EBC-46 are fully understood. Roots are reported as the likely site of biosynthesis for macrocyclic diterpenes (e.g., jatrophane and ingenane) in other taxa of Euphorbiaceae [5], and recent studies suggest a similar situation with *F. picrosperma* with core enzymes encoding for the diterpenoid biosynthetic pathways found to be most highly expressed in roots [17]. A large repertoire of cytochrome P450 genes (P450s) was also reported that showed varied gene expression in both leaf and root tissue, which may correlate with different levels of EBC-46 [11].

The objective of this study was to investigate the potential to identify biomarker genes associated with the biosynthesis of EBC-46, which could be used in selective breeding of *F. picrosperma* to increase yields of the compound from commercial plantations. Towards achieving that, we have used metabolite analysis and differential gene expression approaches.

## 2. Results

### 2.1. Identification of Low and High EBC-46 Producing F. picrosperma Plants

The average EBC-46 concentration of leaf tissue was determined from 12 different plants over a single year. Following HPLC analysis, plants could be separated into low-EBC-46 producers (L1-L5_EBC-46_) and high-EBC-46 producers (H1-H7 _EBC-46_) (Figure 1A,B). The concentration of EBC-46 in leaves of each high-EBC-46-producing plant (H1-7) was statistically and significantly different than in leaves of each low-EBC-46-producing plant (L1-5). We selected 3 L_EBC-46_ (0.008% (L1), 0.008% (L2), and 0.005% (L3) EBC-46 in dried leaves) and 3 H_EBC-46_ (0.051% (H1), 0.050% (H2), and 0.039% (H3) EBC-46 in dried leaves) plants for differential gene expression using quantitative RNA-seq analysis.

### 2.2. Identification and Analysis of Differentially Expressed Genes in Low- versus High-EBC-46-Producing F. picrosperma Plants

Significantly differentially expressed genes (DEGs) within leaf and root tissues of *F. picrosperma* were identified using a log_2_ fold-change of >+/−2 and *p*-value < 0.05 (Figure 2A and Appendix A). Based on these criteria, a total of 246 significant DEGs were identified in leaf tissue of which 145 were upregulated and 101 were downregulated in H_EBC-46_ (Figure 2B). In root tissue, 135 significant DEGs were identified of which 53 were upregulated and 82 were downregulated in H_EBC-46_ (Figure 2B). There was a single common (in leaf and root) upregulated gene, encoding an alpha-glucosidase-like enzyme, while a gene encoding a putative disease-resistance enzyme was the single common downregulated gene. According to gene ontology (GO) analysis of leaf DEGs, there was enrichment of functions involved in transport (65%) and defence (35%), while in root tissue, DEGs were largely enriched in functions associated with metabolomic processes (23%).

### 2.3. Identification and Annotation of Biomarker Genes for High EBC-46 Producing Plants and Validation by RT-qPCR

Candidate biomarker genes for high-EBC-46 production were subsequently investigated from the DEG dataset (Appendix A). In leaf tissue, 10 genes were assessed as candidate biomarkers based on the desired parameters. Of these, eight were more highly expressed in H_EBC-46_ plants, while two were more highly expressed in L_EBC-46_ plants (Figure 3A). The putative disease-resistance protein (*R protein*) and *cytochrome P450 CYP94C1* (*CYP94C1*) genes showed the highest relative abundance in H_EBC-46_ compared to L_EBC-46_ plants (log_2_ fold-change = 10.9 and 6.2, respectively). In root tissue, our DEG analysis had identified six genes as candidate biomarkers; all were more highly expressed in H_EBC-46_ plants (Figure 3B). The 2-alkenal reductase (*2-ARD*) and early response dehydration 7.1 (*ERD-7.1*) genes showed the highest relative abundance in H_EBC-46_ plants (log_2_ fold-change = 9 and 6, respectively).

To investigate the reliability of results obtained from RNA-seq gene expression, those genes with the highest fold-change difference (*CYP94C1*, *R protein*, *2-ARD*, and *ERD-7.1*) were further validated using RT-qPCR on the same plants (Figure 3C,D and Appendix A). The *elongation factor1-α* (*EF1α*), *protein phosphatase 2* (*PP2A*), and *glyceraldehyde 3-phosphate dehydrogenase* (*GAPDH*) were selected as *F. picrosperma* endogenous gene-expression reference genes for normalization in RT-qPCR [18,19]. Following analysis, *PP2A* was identified as the most stable gene in both leaf and root tissue (Appendix A). The results of candidate biomarker gene analysis showed that all genes were highly expressed in H_EBC-46_ plants compared to L_EBC-46_-producing plants, with only *CYP94C1*, *ERD-7.1*, and *2-ARD* being significantly differentially expressed, where coefficient of correlation obtained for the standard curve (R^2^) were 0.691, 0.614, and 0.5949 accordingly.

## 3. Discussion

This study has identified potential biomarker genes for *F. picrosperma* that produce high levels of EBC-46 through metabolite analysis followed by tissue-specific comparative DEG analysis and real-time qPCR validation. The identified DEGs also help to piece together with a potential molecular mechanism for the biosynthesis of EBC-46.

All plants modify their physiology and metabolism in response to external stimuli, such as changes in environmental conditions (e.g., during drought and extreme temperatures) [20], predation, soil nutrient levels, and soil microbial composition [21]. Tissue concentrations of secondary metabolites often vary as a result of these external stimuli. Due to changes in concentrations of natural product, secondary metabolites have previously been reported in medicinally relevant plants, including *Picea abies*, *Quercus robur* L [22], and *Tanacetum vulgare* L. (Asteraceae) [23]. Genes directly involved in their biosynthesis can vary [24], as well as other genes that may not be directly involved in biosynthesis [25]. If the level of gene expression correlates significantly with the tissue concentration of a target secondary metabolite [26], the gene may be a valuable biomarker for selective breeding.

Secondary metabolites often present in only a few tissues, cell types, and organs [12]. In plants, defence metabolites are mostly found in specific tissues or cell types to minimize autotoxicity and/or to maximize the effectiveness of the metabolites [27,28]. In *Nicotiana attenuata*, for example, 63% of non-redundant metabolite spectra showed that secondary metabolites often have organ- and tissue-specific gene expression [29]. This is the case with EBC-46 in *F. picrosperma*, where the fruit contain significantly higher concentrations than other tissues. And whilst the function of EBC-46 in *F. picrosperma* is unclear, it is assumed that it is a deterrent to herbivores, protecting both the fruit and the developing seedling from predation.

RNA-seq DEG analysis was performed on three low- and three high-EBC-46-producing *F. picrosperma* plants, with the aim of identifying gene biomarkers for high levels of EBC-46 production. Gene expression profiling has been widely used to identify candidate genetic biomarkers. For instance, a large transcriptomic dataset in maize (*Zea mays*) allowed for the elucidation of 112 biomarker genes to quantitatively assess the response of maize to changes in soil nitrogen [30]. Our DEG analysis revealed a total of 381 genes significantly up-regulated and down-regulated in root and leaf tissue from H_EBC-46_ plants. Those up-regulated genes were primarily classified into the functional categories of metabolomic processes (in root) and transporter and defence mechanisms (in leaf). In Arabidopsis up-regulated genes are mostly involved in stress tolerance [31], and in wheat, biosynthesis-pathway genes are defined as defence-related molecules [32].

Previous transcriptomic analysis identified the core mevalonate (MVA) and 2-C-methyl-D-erythritol 4-phosphate (MEP) diterpene biosynthesis-pathway enzymes in *Fontainea* [17]. However, based on the comparative quantitative RNA-seq analysis per formed in this study, no MEP or MVA pathway core enzyme genes nor any diterpenoid unit genes were found to be significantly differentially expressed between low- and high-EBC-46 producing plants. Using the selection criterion of annotation, consistency in expression, *p*-value < 0.05, and relative abundance in tissues, 16 DEGs were selected in H_EBC_ and L_EBC_ plants. Four genes (*CYP94C1*, *R protein*, *2-ARD*, and *ERD7.1*) were further selected for RT-qPCR based on highest log_2_ fold-change value, and three genes were validated as being significantly differentially expressed: *CYP94C1* in leaf tissue and *2-ARD* and *ERD7.1* in root tissue.

Whilst *CYP94C1* is a P450 gene, it has not been directly implicated with diterpenoid biosynthesis. In wheat, *CYP94C1* has been associated with defence [33], and in *Arabidopsis thaliana*, it is found in leaf tissue and plays an important role in controlling flowering [34]. *ERD7* has long been known to be related with drought stress [35], and more recently, other stress conditions, such as cold, salt, and excess light, were linked to *ERD7* [36]. However, the molecular role of *ERD7* is still uncertain. The *2-ARD* is known to play an important role in the stress responses in a variety of organisms [37,38], including *Arabidopsis*, where it is involved in disease resistance [39]. In *Jatropha curcas*, which also belongs to the Euphorbiaceae family, *ARD*-like genes encode enzymes that can generate two novel compounds in the diterpenoid biosynthesis pathway [40]. In maize, *2-ARD* significantly increased tolerance to low nitrogen stress [41] and plays a role in the detoxification of reactive carbonyls that generate oxidatively stressed cells in tobacco plants [42]. In *F. picrosperma*, *2-ARD* was expressed in the root and almost exclusively in H_EBC_ plants, whereas in *Artemisia annua*, the expression of *2-ARD* is high in flowers, buds, and leaves and is not detectable in roots [43].

Terpenoids are a class of compounds that are close to steroids. Therefore, they are extensively studied in cancer studies [44]. It is unclear how these DEGs influence the production of EBC-46, nonetheless, they provide a molecular tool that could be used to support breeding programs with a focus on agronomic traits [45]. We envisage that these biomarker genes could be used for rapid screening and classification of individual *F. picrosperma* plants that are high EBC-46 producers. This, in turn, could enable selection and breeding of plant lines that will progressively yield maximal concentrations of EBC-46. Additionally, the aforementioned genes functionally link with stress and defence and support the notion that environmental manipulation, such as changes in temperature, disease, and nutrition, could help drive EBC-46 biosynthesis.

## 4. Materials and Methods

### 4.1. Plant Sample Preparation

Healthy, fully expanded leaves and actively growing root tips, including the apical meristem and root caps, were collected from *Fontainea picrosperma* grown in the University of the Sunshine Coast (Sippy Downs) greenhouse at ambient temperatures (18–24 °C) and relative humidity levels around 80%. Plants were grown in independent pots and kept in the greenhouse at ambient temperatures and humidity, according to Mitu et al. [17]. The harvested plant materials were washed with sterile Milli-Q-filtered water, immediately frozen in liquid nitrogen, and stored at −80 °C until use.

### 4.2. Quantification of EBC-46

Twelve plants (~2–4 years old) with young, healthy, fully expanded leaves were used. Single leaves from each plant were collected between 9.00–10.00 a.m. once a month over a single year. Dried *F. picrosperma* leaves (about 30 mg) were placed in 2 mL safe lock Eppendorf tubes, then frozen in liquid nitrogen, and homogenised by tissue lyser at 30 s for 2 min. For each sample, methanol (0.01667 mL × sample weight) was added and extracted in an ultrasonic bath at room temperature for 3 min. The sample was centrifuged at 8000× *g* for 1 min, and the supernatant was collected for high-performance liquid chromatography (HPLC) analysis. HPLC analyses were performed using an Agilent 1260 system equipped with a diode array detector and the data processed using the Chemstation software (C.01.05). Chromatographic separations were performed with a Halo RP-Amide column (4.6 mm × 150 mm, 2.7 µm) using mobile phases consisting of H_2_O (0.1% formic acid) (A) and CH_3_CN (0.1% formic acid) (B) with the following gradient program: 45–58% B from 0 to 13 min, 58–95% B from 13 to 13.5 min, 95% B from 13.5 to 15 min, 95–45% B from 15–15.1 min, and 45% from 15.1 to 18 min. The detection wavelength was set at 249 nm. The flowrate was 1 mL/min, and the column temperature was set at 40 °C. EBC-46 eluted at 7.3 min, as confirmed by comparison with EBC-46 reference standard provided by QBiotics Ltd.

EBC-46 concentration was determined in leaf tissues of six biological replicates. Student’s *t*-test was used to determine the mean value and statistically significant differences between high and low groups in Microsoft Excel 2016. Values were reported as the mean ± SE from three independent experiments.

### 4.3. Differential Gene Expression Analysis of Leaf and Root of F. picrosperma

Based on the EBC-46 quantitative analysis, three high and three low EBC-46 producers (H_EBC-46_ and L_EBC-46_) were used for RNA sequencing of leaf and root samples (raw reads were deposited into the NCBI, Sequence Read Archive (SRA) database under accession number PRJNA592624). Reads were mapped to the reference *F. picrosperma* transcriptome assembly, previously reported in Mitu et al. [11], derived from leaf and root RNAs. The number of mapped clean reads for each gene was counted and normalized into transcripts per million (TPM) to calculate the expression level of the gene. Differential expression between H_EBC-46_ and L_EBC-46_ samples was established as a log_2_ fold-change using the CLC Genomics workbench (Ver. 11.0.1) set at default parameters (Appendix A). The differential expression significance cut-off was considered as adjusted log2 fold-change value ±2 between H_EBC-46_ and L_EBC-46_ plants and *p*-value < 0.05.

In total, 16 genes were selected based on annotation, consistency in expression, *p*-value < 0.05, and relative abundance in leaf tissue. Next, z-scores were generated for significantly differentially expressed genes based on sequencing depth normalized reads counts. Expression of each gene in leaf and root tissue was analysed by normal clustering. Relative expression profiles of significantly DEGs were presented in the form of a heatmap, which was constructed using Clustvis (https://biit.cs.ut.ee/clustvis/ (accessed on 18 February 2022)) [46], using default parameters and a hierarchical clustering analysis to assess biological sample relatedness. Values were reported as z-scores from three different H_EBC-46_ and L_EBC-46_ plants of *F. picrosperma*.

### 4.4. Reverse Transcription-Quantitative PCR (RT-qPCR)

Approximately 100 mg of leaf and root tissue from 3 H_EBC-46_ and 3 L_EBC-46_ *F. picrosperma* was collected between 9 and 10 a.m. once a month over a year. RNA extraction followed a procedure previously described by Mitu et al. [11]. The RNA integrity was assessed by visualization in 2.0% agarose gels, as well as Nanodrop spectrophotometer 2000c (Thermo Scientific, Waltham, MA, USA) at 260 and 280 nm.

Prior to complementary DNA (cDNA) synthesis, the RNA samples were treated with DNase I reagent kit (Thermo Fisher Scientific) to remove residual genomic DNA. Complementary DNA was then converted from the RNA using the Tetro cDNA synthesis set (Bioline) containing Oligo (dT) primers as well as random primers, and the method directly followed the manufacturer’s instructions. The primer pairs were used to confirm the integrity of RNA and cDNA of all tissue with housekeeping gene elongation factor 1-alpha (*EFT-1α*). Water was used as a negative control. PCRs were prepared using the GoTaq Green Master Mix (Promega), as per the manufacturer’s instructions. Each reaction used the following parameters: initial denaturation cycle at 95 °C for 5 min, subsequent 35 cycles of 15 s at 95 °C, 15 s at 60 °C as the annealing step, 40 s at 72 °C, and a final single extension cycle at 72 °C for 2 min. PCR products were run on 2% Tris-Borate-EDTA (TBE) agarose gel and then visualised with 0.5ug/mL ethidium bromide in a Gel Doc XR+ system coupled with Image Lab Software. Primer sequences for targeted genes, *2-ARD*, *ERD-7.1*, *CYP94C1*, and *R protein*, and reference genes, *EF1α*, *PP2A*, and *GAPDH*, used in this experiment are provided in Appendix A. We first determined the specificity of the designed primer pairs. All PCR products were examined by agarose [1.5% (*w*/*v*)] gel electrophoresis. For each primer pair, the appearance of a single band of the expected size on the agarose gel was considered consistent with the primers being specific to the target gene. RT-qPCR were carried out with a Rotor-Gene^®^ Q (Qiagen) using PowerTrack™ SYBR™ Green Master Mix (Applied Biosystems, Massachusetts, United States). Each 10 μL reaction volume contained 1 μL cDNA, 5 μL PowerTrack^TM^ SYBR^TM^ Green Master Mix, 3.2 μL ultrapure water, and 0.4 μL each primer (400 nM). The reaction conditions included an initial denaturation step of 95 °C for 2 min, followed by 40 cycles of 95 °C for 5 s, 55 °C for 15 s, and 72 °C for 15 s. The dissociation curve analysis was obtained by heating the amplicon from 65 to 95 °C. Each RT-qPCR reaction was performed in three technical replicates and four biological replicates. A triplicate of non-template control was also included for each gene. The optimal annealing temperature was determined by running a qPCR at an annealing temperature gradient from 50 to 60 °C.

To obtain high accuracy of expression stability of reference genes, five different statistical algorithms, i.e., ΔCt [47], geNorm [48], BestKeeper [49], NormFinder [50], and RefFinder [51] were used to evaluate expression stability of the reference genes (Appendix A). RefFinder (https://www.heartcure.com.au/reffinder/?type=reference (accessed on 3 March 2022)) was used to integrate the data obtained from ΔCt method, Genorm, BestKeeper, and NormFinder to calculate the recommended comprehensive ranking order [52]. All the software packages were used according to the manufacturer’s instructions.

### 4.5. Gene Expression and Statistical Analysis

The comparative C_T_ method was used to compare the expression level of each gene [53]. The statistical significance of the differences between EBC-46 concentrations of H_EBC-46_ and L_EBC-46_ plants was determined using the Student’s *t*-test (*p* ≤ 0.05).

## 5. Conclusions and Future Directions

This study has resulted in furthering our knowledge of the molecular mechanisms that influenced EBC-46 production in *F. picrosperma*. We demonstrated that individual *F. picrosperma* plants can produce significantly different concentrations of EBC-46. This difference in concentration was correlated with significant differences in gene expression in both leaf and root tissues. Genes that potentially play a role in metabolomic processes, transporter mechanisms, and defence mechanisms were significantly upregulated in higher EBC-46-producing plants. Following interrogation of sixteen candidate biomarker genes, we have shown that four genes were significantly more highly expressed in higher EBC-46-producing plants, two of which are more highly expressed in leaf tissue, and two in root tissue, with three genes (*CYP94C1*, *2-ARD*, and *ERD7.1*) being significantly differentially expressed and, therefore, being potential biomarkers for EBC-46 concentration in *F. picrosperma*. These results significantly advance our understanding of differentially expressed genes relating to the biosynthesis of EBC-46 in *Fontainea* and may facilitate the development of selected *F. picrosperma* lines with increased levels of EBC-46, as well as the identification of EBC-46 biosynthesis pathway genes.

## Figures and Tables

**Figure 1 molecules-27-03980-f001:**
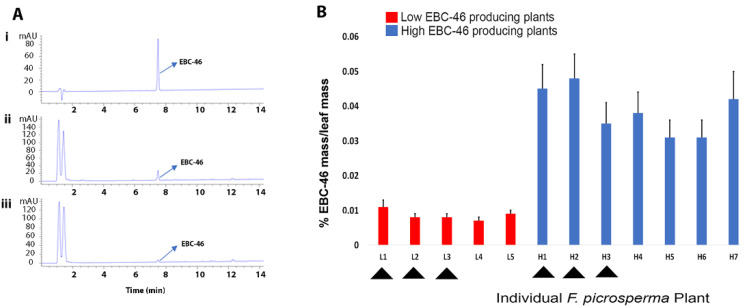
EBC-46 concentration in leaves of 12 different *Fontainea picrosperma* plants. (**A**) HPLC chromatograms at 249nm mAU of (**i**) EBC-46 analytical standard at 0.1 mg/mL (retention time at 7.496 min), (**ii**) leaf extract from an example high-EBC-46-producing *F. picrosperma* plant, and (**iii**) leaf extract from an example low-EBC46-producing *F. picrosperma* plant. (**B**) Average EBC-46 concentration in 12 *Fontainea picrosperma* plants over one year. Arrowheads indicate plants that were selected for differential gene expression analysis. Error bars represent standard error between monthly analyses. L1-5, low-EBC-46-producing plants; H1-7, high-EBC-46-producing plants. The concentration of EBC-46 in leaves of each high-EBC-46-producing plant (H1-7) was statistically and significantly different than in leaves of each low-EBC-46-producing plant (L1-5).

**Figure 2 molecules-27-03980-f002:**
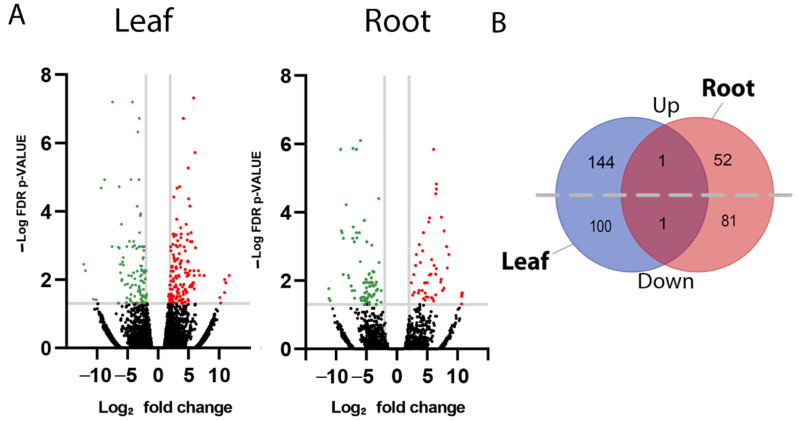
Identification of differentially expressed genes in leaf and root tissue from high- and low-EBC-46-producing *F. picrosperma* plants. (**A**) Volcano plots showing differentially expressed genes. Adjusted *p*-value < 0.05. Red colour represents significantly (+2) upregulated and green colour represents significantly (−2) downregulated genes. (**B**) Venn diagram showing the total number of significantly differentially expressed upregulated (Up) and downregulated (Down) genes in leaf and root tissue, as well as shared genes.

**Figure 3 molecules-27-03980-f003:**
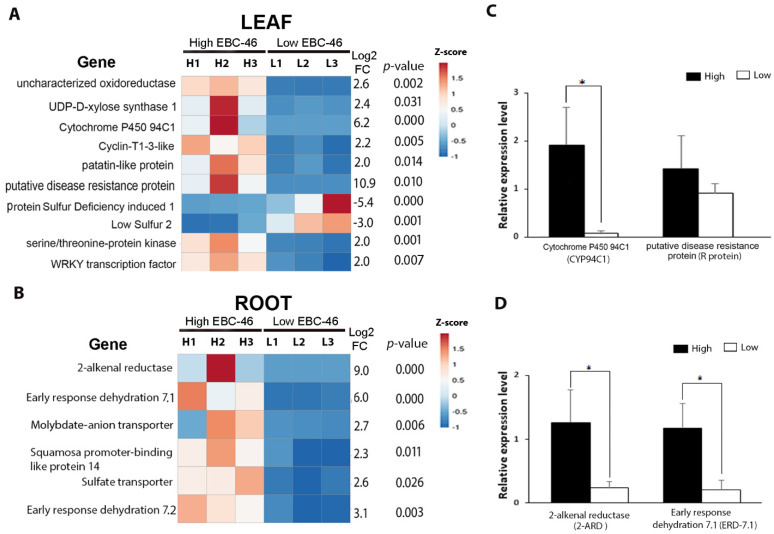
Validation of differentially expressed genes. Heatmaps showing RNA-seq relative gene expression, log2 fold-change, and *p*-value for H_EBC-46_ and L_EBC-46_ *F. picrosperma* in (**A**) leaf tissue and (**B**) root tissue. Four genes were further analysed by RT-qPCR in (**C**) leaf tissue and (**D**) root tissue. * represents statistically significant differences (*p* < 0.05).

## Data Availability

All data generated or analysed during this study are included in this article.

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
