# Peer review of "Identification of Gene Biomarkers for Tigilanol Tiglate Content in *Fontainea picrosperma"

_molecules, 2022, doi:10.3390/molecules27133980_

Round 1
Reviewer 1 Report
Gist: The work by Mitu et al. is relatively straightforwarded with idnetfiication of differnetiallye xpressed genes ( DEGs) and validating them based on log2FC values in a Fontainea. rainforest tree. The EBC-46 yield was checked with comparison of producers based on up/down regulated genes which augments selective breeding purposes.
The introductiona nd discussion need s a major rationale and further discussions respectively. For example, what make EBC interesting , lacunae must be presented. The discussions on DP pathways and their role in treatment of cancers must be deliberated in discussions
The abstract doesn't mention RNA-Seq and gives a subtly confusing statement that qPCR was employed for gene expression.
There must be a pictorial methodology in M & M section
Minor comments but essential
L50: Pl correct to pathways
L130: together WITH a potential
L220: and THE column temperature WAS set
L295: influenceD
L224: statistical
L327: analyzed the data and prepared the data
Author Response
- The introduction and discussion need a major rationale and further discussions respectively. For example, what make EBC interesting, lacunae must be presented. The discussions on DP pathways and their role in treatment of cancers must be deliberated in discussions
Response: We have modified the introduction and discussion to improve the rationale; see lines 31-38, 43-49 and 206-207 in our new version (line references are with track changes open).
- The abstract doesn't mention RNA-Seq and gives a subtly confusing statement that qPCR was employed for gene expression.
Response: Thank you for addressing this point. We mentioned about RNA-seq in our abstract in line 15-17 and 21-22. “Initially, we identified individual plants that were either high (>0.039%) or low EBC-46 (<0.008%) producers, then assessed their differentially expressed genes, within the leaves and roots of these two groups by quantitative RNA sequencing.” “Genes identified as expressed exclusively in either the high or low EBC-46 producing plants were further validated by quantitative PCR.”
- There must be a pictorial methodology in M & M section
Response: Thank you for this suggestion. Our manuscript is focused on identifying biomarkers based on RNA-seq analysis and quantitative analysis (qPCR). To make it convenient, we avoid all graphical presentation of our data analysis.
- Minor comments but essential
Response: Thank you for identifying these points. We have corrected all of these in our new version and added line references in the track change version below.
L50: Pl correct to pathways (line 57)
L130: together WITH a potential (line 143)
L220: and THE column temperature WAS set (line 234)
L295: influenced (309)
L224: statistical (line 238)
L327: analyzed the data and prepared the data (line 339)
Reviewer 2 Report
These work significantly advance our understanding of differential expressed genes relating to the biosynthesis of EBC-46 in Fontainea and may facilitate the development of selected F. picrosperma lines with increased levels of EBC-46, as well as the identification of EBC-46 biosynthesis pathway genes. It is significant for agriculture development and disease treatment.Some questions and suggestions are as follows:
- EBC-46 is not amenable to synthesis on a commercial scale and is currently produced by isolation and purification from the fruit of F. picrosperma. Why look for the DEGs in root and leaf but not in fruit?
- In Fig.1B, the content of EBC-46 in L4-L6 is lower than L1-L3,why not choose the L4-L6?
- Line 83-84:a total of 244 significant DEGs were identified in leaf tissue, of which 145 were upregulated and 101 were downregulated .101+145=244?Also in line 85、86,53+82=133?Also in line 156、157. The number of DEGs must uniform and rigorous.
- In Fig.2A, it is better to make different color to represent the upregulated or downregulated genes.
- The GO enrichment Analysis results are better represented in graphs.
- What are the criteria for selecting candidate gene?I think,firstly, the candidate gene need to be high expression in HEBC-46 but low expression in LEBC-46.So the appear of 2 genes highly expressed in LEBC-46 plants in Fig. 3A is irrational.
- In Fig. 3A,the expression of P450 94C1 in H3 is decrease.so I think the reliability of RNA-seq data need to be verified.
- If we choose the cytochrome P450 94C1 , early response dehydration 7.1 and 2-alkenal reductase as the gene biomakers, further validation between the expression of biomakers and the content of EbC-46 in other unselected F. picrosperma is needed.Otherwise, the evidence is unconvincing.
- The text needs to be checked further.such as,In line 177,no period after [29].
Author Response
EBC-46 is not amenable to synthesis on a commercial scale and is currently produced by isolation and purification from the fruit of F. picrosperma. Why look for the DEGs in root and leaf but not in fruit?
Response: Thank you for highlighting this, which we agree isn’t clearly explained. EBC-46 is present in all tissues of the plant (see Mitu et al. [12]) and you mentioned this point in our updated manuscript, line 47-48. Fruit can be collected only once per year, whereas leaf and root are convenient to collected anytime of the year. However, based on the literature, we anticipated that biosynthesis occurs in the roots and potentially the leaves, not the fruit.
In Fig.1B, the content of EBC-46 in L4-L6 is lower than L1-L3,why not choose the L4-L6?
Response: Yes, we agree. It is a valuable point. However, we selected these plants for RNA-seq based on the EBC-46 data available at the time. We have collected more data since selection, which is presented in Fig 1B, and despite the arbitrary assignment of low and high and changed for some individuals and the ‘order’ of EBC-46 concentration changing between the 12 individuals, statistically each of the L1-L5 plants produce less EBC-46 than H1-H7, which we highlight in line 76-78 “The concentration of EBC-46 in leaves of each high EBC-46 producing plant (H1-7) was statistically significantly different than in leaves of each low EBC-46 producing plant (L1-5)”. Furthermore, there is no statistical difference in production of EBC-46 between L1-L5.
Line 83-84:a total of 244 significant DEGs were identified in leaf tissue, of which 145 were upregulated and 101 were downregulated .101+145=244?Also in line 85、86,53+82=133?Also in line 156 and 157. The number of DEGs must uniform and rigorous.
Response: Thank you for highlighting this point. We have corrected this information in lines 97-100. Also, we have rewritten lines 175 and 176 “Our DEG analysis revealed a total of 381 genes significantly up-regulated and down-regulated in root and leaf tissue from HEBC-46 plants.”. Where we mentioned the total number of DEG genes.
In Fig.2A, it is better to make different color to represent the upregulated or downregulated genes.
Response: Yes, we agree. In our new version we have updated that.
The GO enrichment Analysis results are better represented in graphs.
Response: Thank you for addressing this point. This manuscript focusses on identifying biomarker genes relating to increased EBC-46 production based on differentially expressed data rather than defining the properties of genes. Therefore, we prefer to keep the GO enrichment data a text to avoid unnecessarily diverting the focus of the manuscript.
What are the criteria for selecting candidate gene? I think, firstly, the candidate gene need to be high expression in HEBC-46 but low expression in LEBC-46. So the appear of 2 genes highly expressed in LEBC-46 plants in Fig. 3A is irrational.
Response: Our selection criteria were consistency in expression, P-value <0.05 and relative abundance in leaf tissue in line 253-254. We understand that perspective, but the aim of the project was to look for potential biomarkers, so solely focusing on candidate genes linked with high expression may have excluded relevant candidates.
In Fig. 3A, the expression of P450 94C1 in H3 is decrease.so I think the reliability of RNA-seq data need to be verified.
Response: Yes, we agree with this point. In our study, we have used quantitative analysis (qPCR) to validate the quantitative RNA-seq data (Fig 3C and 3D).
If we choose the cytochrome P450 94C1, early response dehydration 7.1 and 2-alkenal reductase as the gene biomarkers, further validation between the expression of biomakers and the content of Ebc-46 in other unselected F. picrosperma is needed. Otherwise, the evidence is unconvincing.
Response: Thank you for addressing this point. Our qPCR result was based on 4 high and 4 low plants, whereas RNA-seq data presents only 6 (3 high and 3 low). We have added additional information as Supplemental File 3, which presents EBC-46 concentration and qPCR data for all 12 plants with correlation coefficients provided. See lines 135 and 336-337.
The text needs to be checked further. such as, In line 177, no period after [29].
Response: Thank you, we have corrected the point in our updated version.
Reviewer 3 Report
This is an interesting study about potential gene biomarkers associated with biosynthesis of EBC-46 which could be used in selective breeding of F. picrosperma to increase yields of the compound from commercial plantations.
The paper is interesting and the results can help to understand the ways of the development of selected F. picrosperma lines with increased levels of EBC-46, as well as the identification of EBC-46 biosynthesis pathway genes. The abstract presents very clear the objectives of the study. The introduction is supported by well selected bibliographic data. All bibliographic sources are fairly recent and correctly mentioned in text. However, there are some observations:
Line 60-61. I suggest you move this paragraph to Material and Method.
Line 202. Please indicate the value of the ambient temperature and humidity.
I strongly recommend the authors to improve the Introduction and Discussion section. At the Introduction, for example, do not describe the species F. picrosperma at all. In my opinion, is important to briefly describe this species and possibly to improve the references from this point of view.
In the Conclusions section, I suggest you draw some elements from the point of view of future research, if they will be done.
Author Response
Line 60-61. I suggest you move this paragraph to Material and Method.
Response: Yes, we agree and in our recent version we have modified line 72-74 “Average EBC-46 concentration of leaf tissue was determined from 12 different plants over a single year”.
Line 202. Please indicate the value of the ambient temperature and humidity.
Response: The plants are grown in the greenhouse at ambient temperature (18ºC – 24ºC) and relative humidity level around 80%. We have modified the text; accordingly, see line 221-22.
I strongly recommend the authors to improve the Introduction and Discussion section. At the Introduction, for example, do not describe the species picrospermaat all. In my opinion, is important to briefly describe this species and possibly to improve the references from this point of view.
Response: Thank you for this suggestion. We have made several changes to the introduction in alignment with suggestions from the other reviewers and have added additional information about Fontainea picrosperma.
In the Conclusions section, I suggest you draw some elements from the point of view of future research, if they will be done.
Response: Thank you for this suggestion. Our ultimate aim is to identify the biosynthesis pathway for EBC-46 and in our manuscript, we mentioned that these results significantly advance our understanding of differentially expressed genes relating to the biosynthesis of EBC-46 in Fontainea and may facilitate the development of selected F. picrosperma lines with increased levels of EBC-46, as well as the identification of EBC-46 biosynthesis pathway genes, in line 326-329.
Round 2
Reviewer 2 Report
This manuscript basically solved or explained the problem I raised.
Reviewer 3 Report
Dear authors,
The current form of the manuscript has been considerably improved.